# Inverse J-Shaped Relationship of Dietary Carbohydrate Intake with Serum Klotho in NHANES 2007–2016

**DOI:** 10.3390/nu15183956

**Published:** 2023-09-13

**Authors:** Lu Xiang, Mingyang Wu, Yan Wang, Si Liu, Qian Lin, Gang Luo, Lin Xiao

**Affiliations:** Xiangya School of Public Health, Central South University, Changsha 410078, China; xianglu66@csu.edu.cn (L.X.); mingyangwu2016@163.com (M.W.); 226911039@csu.edu.cn (Y.W.); 226911029@csu.edu.cn (S.L.); linqian@csu.edu.cn (Q.L.)

**Keywords:** dietary carbohydrate, serum Klotho, NHANES

## Abstract

Background: The relationship between dietary carbohydrate intake and serum Klotho levels, an aging biomarker, remains uncertain. Objective: This study aimed to investigate the association between dietary carbohydrate intake and serum Klotho levels among American adults aged 40–79. Methods: We analyzed data from 10,669 adults aged 40–79 years who participated in the National Health and Nutrition Examination Survey (NHANES) from 2007 to 2016. Trained interviewers assessed dietary carbohydrate intake using a 24 h dietary recall. Serum Klotho concentrations were measured using commercially available ELISA kits provided by IBL International, Japan, which served as the study outcome. Generalized linear models were used to assess the relationship between the carbohydrate energy percentage and serum Klotho concentration, and restricted cubic spline (RCS) analysis was employed to explore any nonlinear associations. Results: After adjusting for multiple variables, we observed a nonlinear inverse J-shaped relationship (*p* for non-linearity < 0.001) between the carbohydrate energy percentage and serum Klotho levels. Specifically, the highest serum Klotho levels were associated with a total carbohydrate energy percentage ranging from 48.92% to 56.20% (third quartile). When the carbohydrate energy percentage was evaluated in quartiles, serum Klotho levels decreased by 5.37% (95% CI: −7.43%, −3.26%), 2.70% (95% CI: −4.51%, −0.86%), and 2.76% (95% CI: −4.86%, −0.62%) in the first quartile (<41.46%), second quartile (41.46% to 48.92%), and fourth quartile (≥56.20%), respectively, compared to the third quartile. This relationship was more pronounced in male, non-obese and non-diabetic participants under 60 years of age. Conclusion: A non-linear inverse J-shaped relationship exists among the general U.S. middle-aged and older population between the carbohydrate energy percentage and serum Klotho levels, with the highest levels observed at 48.92% to 56.20% carbohydrate intake.

## 1. Introduction

The discovery of the Klotho gene dates back to a pivotal study in 1997, which demonstrated that *Klotho* gene-deficient mice exhibited signs of a shortened lifespan and premature aging, while *Klotho* gene overexpression extended the lifespan of mice [1]. The aging-suppressor protein Klotho, encoded by the Klotho gene, is a transmembrane protein predominantly expressed in the kidneys, brain, and parathyroid glands in three subtypes (α-Klotho, β-Klotho, and γ-Klotho) [2]. This study focused on the α-Klotho protein, which can be released into circulation in soluble forms, hereinafter referred to as Klotho [3]. This protein serves numerous vital biological functions, including regulating phosphorus metabolism, mitigating oxidative stress, reducing inflammation, and modulating energy metabolism. Moreover, it is intricately linked with the regulation of aging [4]. In humans, Klotho protein levels typically exhibit a gradual decline after 40 years [5], a trend further exacerbated by various pathological conditions associated with aging, such as Alzheimer’s disease [6], chronic kidney disease (CKD) [7], and diabetes [8]. Meanwhile, elevated levels of oxidative stress and chronic inflammation levels can suppress Klotho protein expression, thereby affecting its functionality [9,10]. Furthermore, dietary and lifestyle factors have been shown to influence Klotho protein expression [11,12]. Consequently, considering Klotho’s pivotal role in the aging process, targeted modulation of its expression holds promise as a potential avenue for exploring and delaying the aging process.

The relationship between dietary interventions and aging and overall life expectancy has attracted considerable interest in recent years, especially regarding the impact of carbohydrates as a major energy source. However, the association between carbohydrate intake and aging has remained a subject of debate and uncertainty. Some studies have strongly linked high carbohydrate consumption to an increased risk of developing diabetes [8], cardiovascular disease, and age-related diseases [13,14,15]. Conversely, recent investigations have suggested that low-carbohydrate dietary strategies, such as the ketogenic diet [16] and caloric restriction [17], may be associated with prolonged lifespan and improved health. Notably, a prospective cohort study and meta-analysis conducted by Seidelmann et al. [18] revealed a compelling U-shaped association between the energy percentage of carbohydrates and mortality, with low and high carbohydrate levels linked to increased mortality rates. Given the ongoing debate surrounding this topic, additional evidence regarding the association of carbohydrate intake with aging and premature death is warranted.

Klotho, recognized as a key aging marker [2], naturally prompts the hypothesis that carbohydrate intake levels may strongly influence it. However, a comprehensive link between carbohydrate intake levels and serum Klotho remains relatively unexplored in the existing literature. While an epidemiological study has suggested a synchronous increase between carbohydrate intake and soluble Klotho levels in the population [19], experimental research has further supported this finding [20]. Some experimental studies have indicated that a high-sucrose diet can elevate Klotho levels in mice compared to control mice. Conversely, a population-based cross-sectional study found a negative association between carbohydrate consumption and S-Klotho plasma levels in women. However, this relationship was no longer significant after controlling for covariates such as age, lean body mass index, and sedentary time [21]. Hence, further in-depth studies are needed to accurately characterize the association between carbohydrate intake and Klotho levels.

Given the limited existing evidence, the contentious role of carbohydrates in this context, and the public health significance of understanding the relationship between carbohydrate intake and serum Klotho protein levels, we leveraged nationally representative data from the National Health and Nutrition Examination Survey (NHANES) to comprehensively investigate the association between these two factors. This study takes into account the potential for non-linear relationships and aims to contribute to bridging the knowledge gap in our understanding of their connection.

## 2. Materials and Methods

### 2.1. Study Population

The National Health and Nutrition Examination Survey (NHANES) is an epidemiological cross-sectional study designed to collect health and nutrition-related data from the U.S. population. Initiated in 1999, this survey was conducted by the Centers for Disease Control and Prevention (CDC) in the United States using a multi-stage cluster sampling design, with data collection conducted every two years as a recurring cycle. Comprehensive details about this nationwide survey can be openly accessed at https://www.cdc.gov/nchs/nhanes/index.htm on 18 March 2023.

This study pooled data from five consecutive NHANES cycles from 2007 to 2016. The study population was limited to individuals with complete data on serum Klotho, valid dietary recalls, and covariates. Among the initial 50,588 participants aged 40–79 years initially recruited from 2007 to 2016, we first excluded participants lacking serum Klotho data (*n* = 36,824). Subsequently, we further excluded individuals with invalid dietary recalls (*n* = 804) and incomplete covariates (*n* = 1678), extreme caloric intake [22] (<800 or >4200 kcal/day for males and <600 or >3500 kcal/day for females; *n* = 612), as well as subjects with implausibly high carbohydrate intake (≥100% energy intake from carbohydrate, *n* = 1). Ultimately, 10,669 subjects were included in the final analysis, as illustrated in Appendix A. Every participant volunteered to participate in the study and provided written informed consent, which underwent review and approval from the National Center for Health Statistics Institutional Review Board.

### 2.2. Assessment of Dietary Carbohydrate Intake

The dietary interview portion of NHANES utilized the United States Department of Agriculture (USDA) Automated Multiple Pass Method (AMPM) to conduct 24 h dietary recall interviews. These interviews were facilitated by trained interviewers with a computer-assisted dietary interview system, collecting detailed information on all foods and beverages consumed by the participant in the past 24 h (from midnight to midnight). This study estimated the dietary density of carbohydrate intake based on the Food and Nutrition Database for Dietary Studies (FNDDS) provided by the USDA. This was defined as the carbohydrate energy percentage (% of kcal), which was obtained by assuming an energy value of 4 kcal/g for carbohydrates, further converted using macronutrients as a percentage of total energy. The calculation formula is as follows: carbohydrate energy percentage (% of kcal) = carbohydrates intake (g/day) × 4 (kcal/g)/energy intake (kcal/day) [23].

### 2.3. Determination of Serum Klotho Levels

The original frozen serum samples used to measure serum Klotho of participants aged 40–79 years, collected during the NHANES 2007–2016 cycles, were stored at −80 °C. The Klotho concentration analysis was performed by the Northwest Lipid Metabolism and Diabetes Research Laboratory at the University of Washington using a commercially available ELISA kit (IBL International Corporation, Gunma, Japan). All samples were tested in duplicate, and the average of the two measurements was considered the final Klotho concentration. Samples with a difference of more than 10% between duplicate values were re-measured. Quality control samples with values exceeding two standard deviations from the established value were considered invalid for the entire plate and retested. The sensitivity of the Klotho concentration assay was 4.33 pg/mL, and the reference range for Klotho concentration was 285.8–1638.6 pg/mL. For a more detailed description of the Klotho detection method, please refer to the NHANES website [24].

### 2.4. Covariates

The following potential covariates associated with serum Klotho levels based on previous research were considered [25,26,27,28], including age (in continuous), gender (male/female), race (non-Hispanic White/non-Hispanic Black/other Hispanic and Mexican American/other), educational level (<high school/high school/college or above), and income-to-poverty ratio (PIR) obtained from demographic data. Serum cotinine levels (with a serum cotinine cutoff value of 3 ng/mL used to determine smoking status), along with information on alcohol consumption (classified as fewer than 12 drinks per year or more than 12 drinks per year), weight, and height were collected from data obtained at mobile examination centers. Body mass index (BMI, kg/m^2^) was calculated as weight divided by the square of height. Total energy intake (kcal/day) and dietary fiber intake (g/day) were obtained from 24 h dietary recall data. Additionally, we considered several comorbidities (hypertension, diabetes) and the estimated glomerular filtration rate (eGFR, ml/min/1.73 m^2^). Diabetes was defined as meeting one of the following conditions: previously diagnosed with diabetes by a clinician, currently receiving insulin or diabetes medication treatment, HbA1c level ≥6.5%, or fasting glucose level ≥126 mg/dL. Hypertension was defined as meeting one of the following conditions: physician previously diagnosed hypertension, current prescription medication treatment, mean systolic blood pressure ≥140 mmHg, or mean diastolic blood pressure ≥90 mmHg. We used the Chronic Kidney Disease Epidemiology Collaboration equation to assess eGFR [29].

### 2.5. Statistical Analysis

Categorical variables were presented using frequencies (percentages), while normally and skewed distributed continuous variables were described using the mean ± standard deviation (SD) and median (interquartile range), respectively. The continuous Klotho levels were subjected to logarithmic transformation to approximate a normal distribution due to their highly skewed distribution. The carbohydrate energy percentage was evaluated as quartiles and as continuous variable. To examine trends in baseline characteristics across quartiles of the carbohydrate energy percentage, chi-square tests were performed for categorical variables, and linear trend one-way analysis of variance (ANOVA) and Kruskal–Wallis tests were conducted for continuous variables with normal and skewed distributions, respectively.

A multivariate generalized linear regression analysis model was used to estimate the correlation between the carbohydrate energy percentage and Klotho concentration, adjusting for several covariates, including age, gender, BMI, race, education level, PIR, serum cotinine, alcohol drinking, diabetes, hypertension, eGFR, total energy intake, and dietary fiber. To enhance the interpretation of the regression model, we estimated the percentage change in Klotho per 10% increase in the carbohydrate energy percentage. Additionally, the restricted cubic spline (RCS) model was utilized to investigate the non-linear dose–response relationship between the carbohydrate energy percentage and serum Klotho levels in both the overall cohort and the subgroups. Three knots were placed at the 10th, 50th, and 90th percentiles.

Based on previous studies indicating correlations between age [5], gender [30], BMI [31] and diabetes [8] and serum Klotho concentration, we conducted subgroup analyses to explore potential effect modifications within different subgroups based on age (<60 years; ≥60 years), gender (male; female), BMI (non-obese: <30 kg/m^2^; obese: ≥30 kg/m^2^) and diabetes status (no; yes). In each subgroup, we examined the association between the carbohydrate energy percentage and Klotho concentration, treating the median of the carbohydrate energy percentage within each quartile as a continuous variable. A linear trend test was performed to assess its correlation with serum Klotho concentration, while interaction effects were introduced using the Wald test.

Based on the NHANES guidelines, we used weighted estimates with MEC weights. All data collation processes and statistical analyses were conducted using R software (version 4.2.2). A bilateral *p* < 0.05 was considered statistically significant.

## 3. Results

### 3.1. Baseline Characteristics of All Participants

Table 1 presents the baseline characteristics of all participants. A total of 10,669 subjects were included, with an average age of 57.90 ± 10.83 years, of whom 5161 (48.4%) were male. The average carbohydrate intake for all participants was 236.21 ± 96.82 g/day, with an average carbohydrate energy percentage of 48.83 ± 11.24%, and the median serum Klotho concentration (25th–75th percentile) was 801.30 (654.50, 992.00) pg/mL. Significant differences were observed across quartiles of carbohydrate energy percentage for all features except for hypertension and eGFR. Compared to those in the lowest quartile of carbohydrate energy percentage, individuals in the highest quartile were more likely to be older non-obese females with relatively low educational levels and income. They were also less likely to smoke and had lower total energy intake, but had the highest total dietary fiber intake. Notably, these individuals tended to have higher serum Klotho concentrations.

### 3.2. Association between Dietary Carbohydrate Intake and Serum Klotho Levels

The association between the carbohydrate energy percentage and serum Klotho levels is detailed in Table 2. In the fully adjusted model, a significant correlation was observed between the continuous intake of carbohydrate energy percentage and serum Klotho levels. For each 10% increase in the carbohydrate energy percentage, there was a corresponding 1.59% change in serum Klotho levels (95% CI: 0.77%, 2.42%, *p* < 0.001). The carbohydrate energy percentage was included in the regression model as a categorical variable, divided into four quartiles, with the highest percentage change in the Klotho level (third quartile) as the reference. Compared to the third quartile, the decreases in serum Klotho levels for the first quartile, second quartile, and fourth quartile were 5.37% (95% CI: −7.43%, −3.26%), 2.70% (95% CI: −4.51%, −0.86%), and 2.76% (95% CI: −4.86%, −0.62%), respectively. For the relationship between protein, fat intake and serum Klotho levels, see Appendix A. Furthermore, using RCS analysis, a non-linear inverse J-shaped relationship (*p* non-linear < 0.001) was observed between the carbohydrate energy percentage and serum Klotho levels, with an inflection point at 53.71%/kcal (Figure 1). A similar nonlinear dose relationship was observed between fat intake and serum Klotho levels, while protein intake did not display this relationship (see Appendix A). Similarly, the threshold effect analysis showed that before the inflection point, there was a significant positive correlation between the carbohydrate energy percentage (per 10% increment) and serum Klotho levels (percent change = 3.00%, 95% CI: 1.92%, 4.09%, *p* < 0.001) (Table 3). However, after the inflection point, the correlation between the carbohydrate energy percentage (per 10% increment) and serum Klotho levels decreased and exhibited no statistical significance (percent change = 1.70%, 95% CI: −3.53%, 0.16%, *p* = 0.078).

### 3.3. Subgroup Analysis

In the stratified analyses, age, sex, BMI and diabetes did not significantly modify the association between the carbohydrate energy percentage and serum Klotho levels (all *p*-interactions > 0.05). After adjusting for potential confounders, a significant association between the carbohydrate energy percentage and serum Klotho levels was observed in participants who were younger than 60 years old, male, non-obese, and did not have a history of diabetes. (Table 4). With each 10% increase in the carbohydrate energy percentage, serum Klotho concentrations significantly varied in younger individuals (percent change = 2.07%, 95% CI: 0.99%, 3.17%), males (percent change = 1.86%, 95% CI: 0.79%, 2.94%), non-obese individuals (percent change = 2.00%, 95% CI: 0.84%, 3.17%) and those without diabetes (percent change = 1.64%, 95% CI: 0.72%, 2.58%).

In addition, the RCS was used to estimate the dose–response relationship between the carbohydrate energy percentage and serum Klotho levels in each subgroup (see Appendix A). The results show a significant nonlinear dose–response relationship between the carbohydrate energy percentage and serum Klotho levels in the elderly (*p* for non-linearity = 0.018) and middle-aged groups (*p* for non-linearity = 0.002), as well as in non-obese (*p* for non-linearity = 0.002) and obese individuals (*p* for non-linearity = 0.013), both non-diabetic (*p* for non-linearity = 0.001) and diabetic (*p* for non-linearity = 0.021), and in females (*p* for non-linearity < 0.001). In contrast, there was a linear dose–response relationship between the two among male participants (*p* for non-linearity = 0.056).

## 4. Discussion

In our study based on a US national cohort study, we observed an inverted J-shaped association between the carbohydrate energy percentage and serum Klotho levels, with an inflection point at approximately 53.71% and the highest levels of serum Klotho at 48.92% to 56.20% of carbohydrate energy percentage, even after further adjusting for confounding factors. Furthermore, this association was more prominent in males, non-obese individuals, and those without diabetes who were under 60 years of age.

Research on the impact of dietary carbohydrate intake on serum Klotho levels in the general population is currently limited, and the reported results are inconsistent. For instance, Wu et al. [32] showed no association between Klotho and both a low-carbohydrate diet and low-carbohydrate diet scores (n = 7906). However, in the FIT-AGEING study (n = 72) [21], De-la-O et al. found a negative correlation between carbohydrate intake and serum Klotho levels in females, but this correlation disappeared after controlling for covariates such as age, lean body mass index, and sedentary time. In contrast, Ostojic et al. [19] reported a positive association between higher carbohydrate intake and soluble Klotho levels after adjusting for age and gender (n = 2637). Overall, these studies suggest that no consensus has been reached on the relationship between carbohydrate intake and Klotho levels. The discrepancies in findings may partly stem from differences in the target population, sample size, confounding factors, and sampling weights. Moreover, previous research on the association between carbohydrate intake and serum Klotho levels has often focused on describing linear relationships, potentially obscuring real information provided by continuous carbohydrate intake data. This study provides novel insights by observing the sustained association between dietary carbohydrate intake and serum Klotho levels, along with a series of subgroup analyses.

Our findings reveal an interesting relationship between carbohydrate intake and Klotho levels, and this association seems to be more pronounced in males, non-obese individuals, and those under 60 years of age without diabetes. Based on the inverted J-shaped dose–response curve of the present study, it was first observed that serum Klotho levels would increase substantially with increasing carbohydrate intake among subjects with carbohydrate energy percentage below 53.71%. We hypothesize that several mechanisms may underlie these observed relationships. Dietary carbohydrates themselves can serve as a source of certain vitamins [33,34], minerals [35], and fiber [27], which could positively influence Klotho protein expression. Additionally, low-carbohydrate-intake diets tend to inevitably reduce the intake of high-quality carbohydrates [28,36], including fruits, non-starchy vegetables, and whole grains, and simultaneously alter the amount and type of gut microbiota and affect hepatic FGF21 production [20,37], all of which may downregulate Klotho levels by modulating the levels of inflammation and oxidative stress. Given these insights, extremely low dietary carbohydrate intake is not conducive to elevating serum Klotho levels to counteract the aging process. However, further population studies in the future are still needed to corroborate our conclusions.

Furthermore, our study showed that serum Klotho levels will slightly decrease with increasing carbohydrate intake among subjects with a carbohydrate energy percentage above 53.71%, although this association lacks statistical significance. From a biological perspective, excessive carbohydrate intake has been widely acknowledged to induce a variety of adverse physiological effects, including the accumulation of advanced glycation end-products [38,39], insulin resistance [3], the suppression of adiponectin gene expression resulting in disrupted lipid metabolism, and the stimulation of oxidative stress and inflammatory processes [40]. These factors have been demonstrated to downregulate Klotho expression. Nevertheless, further research is required to validate our findings and explore potential underlying mechanisms.

Our subgroup analysis results revealed a significant correlation between the carbohydrate energy percentage and serum Klotho levels in middle-aged participants, but this association was not observed in the elderly participants. This finding may be attributed to the fact that Klotho, as a longevity factor, naturally declines with age after 40 years [5]. Thus, Klotho levels might remain relatively high in middle age, making the link between carbohydrate intake and Klotho levels more apparent. In addition, individuals aged 60 and above might experience higher levels of systemic inflammation, oxidative stress, and reduced glucose tolerance, which may contribute to the development of hypertension, diabetes, and cardiovascular disease [41,42,43]. These physiological and disease-related conditions can directly or indirectly influence Klotho expression, thereby obscuring the potential correlation between carbohydrate intake and Klotho levels. Furthermore, we also observed disparities in the association between dietary carbohydrate intake and serum Klotho levels among males and females, with a more pronounced correlation in males. One potential explanation for these gender-specific findings is that testosterone levels in males might upregulate the expression of the *Klotho* gene [44,45,46]. However, future research is warranted to evaluate these variations in the association between dietary carbohydrate intake and Klotho levels within gender categories. Consistent with the literature, this study also revealed that the association may vary according to the BMI. In this respect, the correlation between carbohydrate intake and serum Klotho levels appeared more significant in non-obese individuals than obese individuals. Plausible explanations for this discrepancy could be that the excess visceral adipose tissue in obese subjects leads to chronic inflammation [25], which consequently lowers serum Klotho levels, or leads to the need to consume higher carbohydrate intake to maintain resting state energy expenditure, thus masking the true association between carbohydrate intake and Klotho levels. Furthermore, the relationship between carbohydrate intake and serum Klotho levels was weakened in the context of diabetes. This is consistent with previous studies that showed that serum Klotho levels decrease significantly in diabetic patients, but not in those without diabetes [47,48,49]. Despite these explanations, the difference between subgroups should be interpreted with caution, as the interaction between subgroups did not reach statistical significance.

In addition to the contribution of carbohydrates, balanced nutrition should not be overlooked. Carbohydrates do not independently influence the anti-aging process, but rather interact with the other two macronutrients: proteins and fats. A dietary pattern, known as the “Okinawan ratio”, characterized by a carbohydrate-to-protein ratio of 9:1, has attracted attention for its contribution to longevity [50]. This model was derived from a study of residents of Okinawa Prefecture in southern Japan, who on average live longer and have a lower risk of aging-related diseases [51]. Wahl et al. [52] showed that low-protein and high-carbohydrate diets could positively affect brain aging in mice, with the carbohydrate-to-protein ratio closely aligning with the Okinawan ratio. Protein plays a crucial role in these processes, prompting us to investigate its relationship with carbohydrate intake and serum Klotho level. We observed a positive correlation between protein intake and increased serum Klotho levels, similar to Shafie et al. [53], who reported that a high-protein diet elevated α-Klotho levels in aged rats. However, a cross-sectional study found an inverse association between protein intake and serum Klotho levels in sedentary middle-aged women [21]. The association between protein intake and Klotho warrants further investigation. Furthermore, we observed a significant nonlinear relationship between fat intake and serum Klotho levels. Our study showed that serum Klotho levels would increase substantially with increasing fat intake, in contrast to the findings of Wu et al. [32]. In their study, the low-fat diet revealed a null association with s-Klotho levels. Conversely, we showed that excessive fat intake resulted in a noticeable decrease in serum Klotho levels, which is consistent with animal studies [3,54,55]. Consequently, further investigation is warranted to elucidate the optimal balance among carbohydrates, proteins, and fats and their influence on serum Klotho levels.

Overall, the association between carbohydrate intake and Klotho levels was nonlinear, resembling the U-shaped association between the carbohydrate energy percentage and life expectancy reported in previous studies, with the longest lifespan observed at a carbohydrate energy percentage of 50–55% [18]. Our study consistently showed that both excessively low and high carbohydrate intake reduced Klotho levels, with the highest Klotho levels observed at a carbohydrate energy percentage ranging from 48.92% to 56.20%. Therefore, the association between carbohydrate diets and organismal aging should be interpreted with caution. Moderate carbohydrate consumption may prove more advantageous in postponing aging when compared to extremely high or low intake. However, due to the limited research on this subject so far, it is imperative to explore the exact mechanisms governing the ideal percentage of carbohydrate energy and its connection with Klotho. Future comprehensive investigations into the role of Klotho protein in aging and associated ailments, as well as the influence of carbohydrate diets on Klotho expression, will contribute to the development of more effective strategies for anti-aging and overall health.

Several limitations of our study should be acknowledged. First, despite the observed significant association between carbohydrate intake and serum Klotho levels, the cross-sectional design of the NHANES study prevented us from establishing causal or temporal associations. Second, the inclusion of serum Klotho data only for participants aged 40–79 years in the NHANES database may limit the generalizability of our findings to other age groups. Third, despite adjusting for multiple covariates in our statistical analyses, the possibility of unmeasured or unrecorded confounders influencing our results cannot be ruled out. Finally, the study was limited by the inability to specify the type of carbohydrate consumed by the study population, which hinders an in-depth understanding of the relationship between carbohydrates intake and serum Klotho levels. Therefore, it is essential to interpret the results cautiously.

## 5. Conclusions

In summary, this study found an inverted J-shaped association between the carbohydrate energy percentage and serum Klotho levels, with an inflection point at about 53.71% and the highest levels observed at 48.92% to 56.20% carbohydrate intake. Our findings emphasize the possibility of maintaining an optimal carbohydrate intake to prevent aging in the middle-aged and elderly groups.

## Figures and Tables

**Figure 1 nutrients-15-03956-f001:**
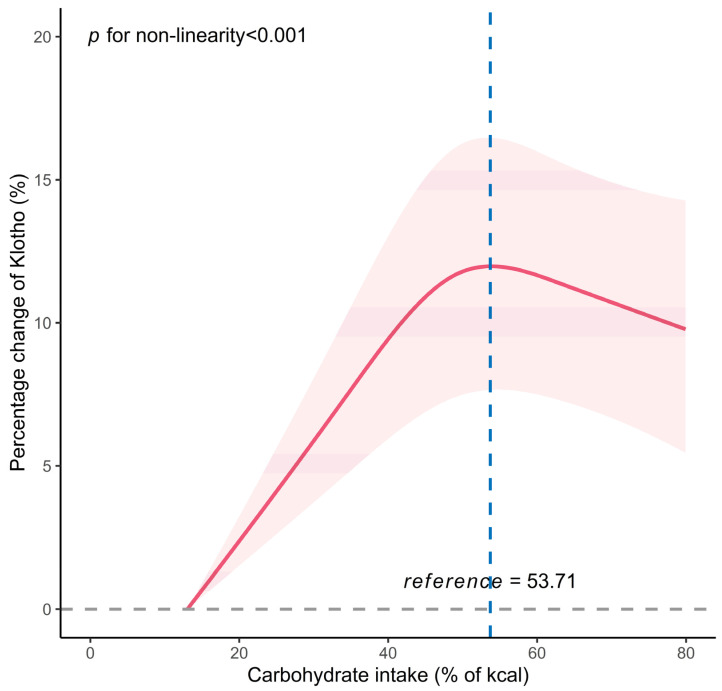
The dose–response relationship between the carbohydrate energy percentage and serum Klotho level among 10,669 participants. The restricted cubic spline curve was created out with 3 knots at the 10th, 50th, and 90th percentiles of carbohydrate energy percentage. The reference point is the maximum percentage change in Klotho. The solid line represents point estimation on the association of the carbohydrate energy percentage with serum Klotho, and the shaded portion represents 95% CI estimation. The model was adjusted for age, sex, BMI, PIR, education attainment, ethnicity, serum cotinine, alcohol drinking, diabetes, hypertension, eGFR, dietary energy intake, and dietary fiber.

**Table 1 nutrients-15-03956-t001:** Basic characteristics of NHANES participants (N = 10,669) *.

Characteristics	All	Quartiles of Carbohydrate Intake, % of kcal				
		Q1 (<41.46)	Q2 (41.46 to 48.92)	Q3 (48.92 to 56.20)	Q4 (≥56.20)	*p* Value
N	10,669	2667	2667	2667	2668	
Age, years	57.90 ± 10.83	57.54 ± 10.52	57.48 ± 10.69	58.21 ± 10.99	58.38 ± 11.10	0.002
Sex, *n* (%)						<0.001
male	5161 (48.4)	1445 (54.2)	1357 (50.9)	1227 (46.0)	1132 (42.4)	
female	5508 (51.6)	1222 (45.8)	1310 (49.1)	1440 (54.0)	1536 (57.6)	
BMI, kg/m^2^						0.013
<30 kg/m^2^	6164 (57.8)	1499 (56.2)	1504 (56.4)	1556 (58.3)	1605 (60.2)	
≥30 kg/m^2^	4505 (42.2)	1168 (43.8)	1163 (43.6)	1111 (41.7)	1063 (39.8)	
Race, *n* (%)						<0.001
Non-Hispanic White	4947 (46.4)	1407 (52.8)	1301 (48.7)	1179 (44.2)	1060 (39.7)	
Non-Hispanic Black	2076 (19.4)	542 (20.3)	511 (19.2)	525 (19.7)	498 (18.7)	
Other Hispanic	1141 (10.7)	211 (7.9)	263 (9.9)	326 (12.2)	341 (12.8)	
Mexican American or Other	2505 (23.5)	507 (19.0)	592 (22.2)	637 (23.9)	769 (28.8)	
Education, *n* (%)						<0.001
<High school	2752 (25.8)	575 (21.6)	632 (23.7)	698 (26.2)	847 (31.7)	
High school	2379 (22.3)	600 (22.5)	617 (23.1)	607 (22.8)	555 (20.8)	
College or above	5538 (51.9)	1492 (55.9)	1418 (53.2)	1362 (51.0)	1266 (47.5)	
PIR	2.67 ± 1.65	2.90 ± 1.67	2.79 ± 1.65	2.59 ± 1.63	2.40 ± 1.61	<0.001
Serum cotinine, ng/mL	0.03 (0.01, 1.49)	0.04 (0.01, 46.25)	0.03 (0.01, 1.31)	0.03 (0.01, 0.56)	0.03 (0.01, 0.44)	<0.001
Alcohol drinking, *n* (%)						<0.001
More than 12 drinks/year	3041 (28.5)	470 (17.6)	704 (26.4)	840 (31.5)	1027 (38.5)	
Less than 12 drinks/year	7628 (71.5)	2197 (82.4)	1963 (73.6)	1827 (68.5)	1641 (61.5)	
Diabetes, *n* (%)						<0.001
No	8110 (76.0)	1961 (73.5)	2010 (75.4)	2046 (76.7)	2093 (78.4)	
Yes	2559 (24.0)	706 (26.5)	657 (24.6)	621 (23.3)	575 (21.6)	
Hypertension, *n* (%)						0.053
No	4855 (45.5)	1158 (43.4)	1209 (45.3)	1235 (46.3)	1253 (47.0)	
Yes	5814 (54.5)	1509 (56.6)	1458 (54.7)	1432 (53.7)	1415 (53.0)	
eGFR, mL/min/1.73 m^2^	83.95 ± 19.61	83.30 ± 19.58	83.89 ± 19.24	84.09 ± 19.44	84.53 ± 20.19	0.148
Dietary intake						
Carbohydrate, g/day	236.21 ± 96.82	178.72 ± 72.19	233.52 ± 85.10	255.14 ± 92.93	277.43 ± 105.11	<0.001
Energy, kcal/day	1961.27 ± 732.44	2065.11 ± 756.50	2059.55 ± 746.67	1945.99 ± 706.36	1774.51 ± 679.81	<0.001
Carbohydrate intake, % of kcal	48.83 ± 11.24	34.53 ± 5.73	45.38 ± 2.14	52.47 ± 2.09	62.94 ± 5.93	<0.001
Fat intake, % of kcal	33.77 ± 9.20	41.11 ± 9.58	36.55 ± 6.50	32.46 ± 5.40	24.96 ± 5.91	<0.001
Protein intake, % of kcal	15.97 ± 5.00	18.55 ± 5.92	16.47 ± 4.45	15.28 ± 4.11	13.57 ± 3.89	<0.001
Fiber, g/day	16.66 ± 9.78	13.85 ± 8.21	16.52 ± 8.88	17.72 ± 9.84	18.54 ± 11.28	<0.001
Serum Klotho, pg/mL	801.30 (654.50, 992.00)	769.70 (631.35, 956.55)	799.30 (657.80, 991.40)	820.70 (666.95, 1014.25)	814.55 (656.85, 1007.32)	<0.001

* Values are presented as the mean ± SD, median (interquartile range), or *n* (%). Abbreviations: BMI, body mass index; PIR, family poverty income ratio; eGFR, estimated glomerular filtration rate.

**Table 2 nutrients-15-03956-t002:** The association between the carbohydrate energy percentage and serum Klotho levels among all participants.

Carbohydrate Intake, % of kcal	N	Crude Model		Adjusted Model *	
		Percent Changes (%) and 95% CI	*p* Value	Percent Changes (%) and 95% CI	*p* Value
Per 10% increases		2.17 (1.37, 2.98)	<0.001	1.59 (0.77, 2.42)	<0.001
Q1 (<41.46)	2667	−6.63 (−8.66, −4.55)	<0.001	−5.37 (−7.43, −3.26)	<0.001
Q2 (41.46 to 48.92)	2667	−3.33 (−5.15, −1.47)	<0.001	−2.70 (−4.51, −0.86)	0.006
Q3 (48.92 to 56.20)	2667	Ref	Ref	Ref	Ref
Q4 (≥56.20)	2668	−2.33 (−4.46, −0.16)	0.039	−2.76 (−4.86, −0.62)	0.014

* Adjusted for age, gender, BMI, race, education level, PIR, serum cotinine, alcohol drinking, diabetes, hypertension, eGFR, dietary energy intake and dietary fiber. Abbreviations: CI, confidence interval.

**Table 3 nutrients-15-03956-t003:** Threshold analyses of dietary carbohydrate intake (per 10% increment) on the level of Klotho using two-piecewise regression models.

Carbohydrate Intake, % of kcal	Crude Model		Adjusted Model *	
	Percent Changes (%) and 95% CI	*p* Value	Percent Changes (%) and 95% CI	*p* Value
Fitting by two-piecewise linear model				
<53.71	3.67 (2.64, 4.71)	<0.001	3.00 (1.92, 4.09)	<0.001
≥53.71	−1.34 (−3.19, 0.55)	0.167	−1.70 (−3.53, 0.16)	0.078

* Adjusted for age, gender, BMI, race, education level, PIR, serum cotinine, alcohol drinking, diabetes, hypertension, eGFR, dietary energy intake, and dietary fiber.

**Table 4 nutrients-15-03956-t004:** The relationship between the carbohydrate energy percentage and serum Klotho levels, stratified by age, sex, BMI and diabetes.

Participants	Carbohydrate Intake, % of kcal	Percent Changes (%) and 95% CI	*p* Value	* *p* for Interaction
Age, years				0.517
<60	Per 10% increases	2.07 (0.99, 3.17)	<0.001	
	Q1 (<41.46)	−7.21 (−9.81, −4.54)	<0.001	
	Q2 (41.46 to 48.92)	−3.49(−5.97, −0.94)	0.01	
	Q3 (48.92 to 56.20)	Ref	Ref	
	Q4 (≥56.20)	−3.53 (−6.51, −0.46)	0.028	
≥60	Per 10% increases	0.92 (−0.47, 2.33)	0.198	
	Q1 (<41.46)	−2.47 (−6.01, 1.19)	0.189	
	Q2 (41.46 to 48.92)	−1.84 (−5.22, 1.66)	0.302	
	Q3 (48.92 to 56.20)	Ref	Ref	
	Q4 (≥56.20)	−1.38 (−5.22, 2.60)	0.493	
Sex				0.619
male	Per 10% increases	1.86 (0.79, 2.94)	0.001	
	Q1 (<41.46)	−6.65 (−9.60, −3.60)	<0.001	
	Q2 (41.46 to 48.92)	−5.09 (−7.57, −2.54)	<0.001	
	Q3 (48.92 to 56.20)	Ref	Ref	
	Q4 (≥56.20)	−3.56 (−7.45, 0.48)	0.089	
female	Per 10% increases	1.49 (0.46, 2.53)	0.006	
	Q1 (<41.46)	−4.68(−7.72, −1.54)	0.005	
	Q2 (41.46 to 48.92)	−0.74 (−3.77, 2.39)	0.642	
	Q3 (48.92 to 56.20)	Ref	Ref	
	Q4 (≥56.20)	−2.20 (−5.39, 1.10)	0.194	
BMI, kg/m^2^				0.079
<30	Per 10% increases	2.00 (0.84, 3.17)	0.001	
	Q1 (<41.46)	−7.64 (−10.45, −4.75)	<0.001	
	Q2 (41.46 to 48.92)	−4.01 (−6.52, −1.43)	0.004	
	Q3 (48.92 to 56.20)	Ref	Ref	
	Q4 (≥56.20)	−3.90 (−6.66, −1.05)	0.01	
≥30	Per 10% increases	1.04 (−0.02, 2.11)	0.059	
	Q1 (<41.46)	−2.01 (−5.62, 1.74)	0.295	
	Q2 (41.46 to 48.92)	−0.54 (−3.96, 3.00)	0.762	
	Q3 (48.92 to 56.20)	Ref	Ref	
	Q4 (≥56.20)	−0.89 (−4.39, 2.73)	0.625	
Diabetes				0.336
No	Per 10% increases	1.64 (0.72, 2.58)	<0.001	
	Q1 (<41.46)	−6.06 (−8.18, −3.89)	<0.001	
	Q2 (41.46 to 48.92)	−2.80 (−4.88, −0.66)	0.013	
	Q3 (48.92 to 56.20)	Ref	Ref	
	Q4 (≥56.20)	−2.91 (−5.33, −0.43)	0.025	
Yes	Per 10% increases	1.26 (−0.45, 3.01)	0.154	
	Q1 (<41.46)	−2.42 (−7.17, 2.57)	0.339	
	Q2 (41.46 to 48.92)	−1.99 (−6.53, 2.76)	0.408	
	Q3 (48.92 to 56.20)	Ref	Ref	
	Q4 (≥56.20)	−2.39 (−6.85, 2.29)	0.316	

* *p* value for interaction of carbohydrate intake with age, sex, BMI or diabetes. Abbreviations: CI, confidence interval; BMI, body mass index.

## Data Availability

The datasets produced and examined during the current study can be openly accessed via the NHANES website (https://wwwn.cdc.gov/nchs/nhanes/Default.aspx, visited on 18 March 2023).

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
