# Peer review of "Inverse J-Shaped Relationship of Dietary Carbohydrate Intake with Serum Klotho in NHANES 2007–2016"

_nutrients, 2023, doi:10.3390/nu15183956_

Round 1
Reviewer 1 Report
Dear authors
This manuscript shows important data of statistically non-linear correlation between carbohydrate intake ratio and serum concentration of Klotho, an anti-aging factor. This study is very interesting, and it can provide an important contribution to “Nutrition” and “Aging” research.
This manuscript requires a slight revision to be acceptable for the Nutrites journal. I show the detail comment below.
Major
1.There are several experimental reports on aging and carbohydrates intake (e.g. doi: 10.1016/j.celrep.2018.10.070.) Please compare and discuss these reports with this study.
2.Is there a reason you didn't check other aging biomarkers? In particular, IGF1 is related to carbohydrate metabolism and is known interaction with Klotho and aging (e.g. DOI: 10.3390/cells10061376; 10.1530/JOE-13-0285). Citation of appropriate literature, if available, may be substituted.
3.Are similar analyzes underway for changes in Klotho with other nutrients, fat and protein intake? Citation of appropriate literature, if available, may be substituted.
Minor
1.The author said ”this association was more prominent in older and non-obese male participants.”(page 9, line 272)in the first paragraph of Discussion . However, the data of Tableï¼” showed <60 is more pronounced than 60≤. Please make sure.
I recommend the check by Native-English speaker before resubmission.
For example, I think that a sentence break in the first paragraph of "Discussion" is unclear and grammatically incorect.
Reviewer 2 Report
In this article, authors analyzed public databases to relate carbohydrate intake with Khloto serum levels. The database provides the data and the authors just analyzed them.
In the selection of subjects, an important point is that almost 24 % are already diabetic type II and this fact clearly could bias the conclusion of the study. Therefore, the authors should analyze the data using two different groups: Diabetic and non-diabetic.
Furthermore, The effects of a high carbohydrate diet depend on the nature of the carbohydrates (slow and rapid digestible or the glycemic index) and the combination with a high-fat diet. All these aspects are missing in the results and need to be included in the study before any conclusion can be reached.
Also, it would be nice to have some data regarding insulin levels and insulin sensitivity (HOMA) in the study.
Therefore, all these aspects should need to be incorporated in the article.
Fine, just a minor revision.
Round 2
Reviewer 2 Report
The authors have significantly improved the manuscript including the different analyses suggested by the reviewers.
Fine